# Investigation of Optic Nerve Sheath Diameter, Intraocular Pressure, and Dry Eye in Patients with Borderline Personality Disorder: The Role of Childhood Trauma

**DOI:** 10.3390/jcm14165886

**Published:** 2025-08-20

**Authors:** Tunahan Sun, Demet Dursun Çakar, Caner Yeşiloğlu, Mehmet Emin Demirkol, Lut Tamam, Kerim Uğur, Hatice Polat

**Affiliations:** 1Department of Psychiatry, Düziçi State Hospital, Osmaniye 80600, Turkey; 2Department of Ophthalmology, Düziçi State Hospital, Osmaniye 80600, Turkey; demetdursun@hotmail.com; 3Department of Psychiatry, Faculty of Medicine, Cukurova University, Adana 01100, Turkey; yesiloglucaner@gmail.com (C.Y.); emindemirkol@gmail.com (M.E.D.); ltamam@gmail.com (L.T.); 4Department of Psychiatry, Faculty of Medicine, Malatya Turgut Özal University, Malatya 44000, Turkey; premirek@gmail.com; 5Faculty of Health Sciences, Malatya Turgut Özal University, Malatya 44000, Turkey; hatice_ce.polat@hotmail.com

**Keywords:** borderline personality disorder, optic nerve sheath diameter, intraocular pressure, dry eye, childhood trauma, indicators of neurobiological changes, biomarkers

## Abstract

**Background/Objectives**: Borderline personality disorder (BPD) is a psychiatric disorder characterized by emotional instability, impulsive behavior, and impaired interpersonal relationships. It is associated with a high prevalence of childhood trauma and neurobiological changes. This study aimed to compare ophthalmologic parameters, namely, optic nerve sheath diameter, intraocular pressure, and dry eye, in patients with BPD with healthy controls and to investigate the relations between these parameters and childhood trauma. **Methods**: This study included 51 female patients with BPD between the ages of 18 and 35 years, who were not using psychotropic medication, and 51 healthy controls matched for age and educational level. Optic nerve sheath diameter, intraocular pressure, and tear break-up time were measured, and trauma history was evaluated using the Childhood Trauma Questionnaire-Short Form. Independent *t*-test and Pearson correlation analysis were used in statistical analyses. **Results**: Patients with BPD were found to have significantly higher mean optic nerve sheath diameter scores (left: 3.94 ± 0.43, right: 3.97 ± 0.47) compared with healthy controls (left: 3.76 ± 0.44, right: 3.78 ± 0.45) (*p* < 0.05). The groups showed no significant difference in intraocular pressure and dry eye parameters (*p* > 0.05). A significant positive correlation was noted between emotional abuse scores and the optic nerve sheath diameter of the left eye in patients with BPD (*p* < 0.05; r = 0.364). **Conclusions**: An increased optic nerve sheath diameter may be a potential peripheral biomarker reflecting chronic stress or changes in intracranial physiology in patients with BPD. This increase is particularly associated with a history of emotional abuse. Ophthalmological parameters may contribute to understanding the neurobiological basis of BPD and serve as peripheral biomarkers or indicators of neurobiological changes.

## 1. Introduction

Borderline personality disorder (BPD) is a psychiatric disorder characterized by identity confusion, unstable interpersonal relationships, sudden mood swings, impulsive behavior, periodic intense anger, persistent feelings of emptiness, suicidal and self-mutilative behavior, temporary and stress-related paranoid thoughts, and severe dissociative symptoms (e.g., depersonalization and derealization) [1]. Its prevalence in adults varies between 0.7% and 2.7% [2,3]. This rate reaches 10% among all psychiatry outpatients and 15–20% among inpatients. BPD causes serious functional impairment and intensive use of health care services. It is also characterized by a significantly higher suicide rate compared with the general population [4]. The past decade has seen an increase in the number of biological studies conducted to elucidate the etiology of BPD [5,6].

Ophthalmological biomarkers and indicators of neurobiological changes in psychiatric disorders appear to be rather overlooked despite their clinical significance. However, psychiatric disorders have numerous ophthalmological presentations that may be clinically significant for patients. In psychiatric disorders, such as schizophrenia and mood disorders, researchers have investigated the use of retinal biomarkers to shed light on the pathophysiological mechanisms underlying functional abnormalities in the brain. For example, schizophrenia has been associated with retinal thinning and a decrease in macular volume and thickness. In bipolar disorder, thinning has been shown in the peripapillary retinal nerve fiber layer and, with disease progression, in the inner plexiform layer and ganglion cell layer. In addition, decreased rod b-wave amplitude and delayed cone b-wave latency have been observed in patients diagnosed with bipolar disorder at high genetic risk. In depressive disorder, prolonged b-waves have been found at the cone level [7,8]. Patients with BPD, meanwhile, have shown a lower thickness of the peripapillary retinal fiber layer and central retina compared with healthy controls [9].

Childhood trauma (CT) is associated with the emergence of systemic diseases and psychiatric disorders affecting multiple systems, such as the immune, endocrine, and nervous systems [10,11]. For example, adults with a history of multiple CTs have a higher risk of developing psychiatric disorders, such as anxiety, mood, personality, eating, and post-traumatic stress disorder [12,13]. From 30% to 90% of patients with BPD experienced trauma in childhood, which is significantly higher than the rates recorded in other personality and psychiatric disorders [4].

Stress response dysregulation may also cause retinal changes through inflammation, and individuals who experienced adverse life events in childhood are more likely to develop such changes [11]. Some specific CT types, such as physical abuse, have been associated with prolonged latencies of rod and cone cells of the retina in young girls at familial risk for schizophrenia and mood disorders [10]. However, to the best of our knowledge, the relationship between CT and other ophthalmologic parameters has not been investigated despite CT’s higher prevalence compared with other psychiatric disorders in patients with BPD.

The first of these parameters is the optic nerve sheath diameter (ONSD), which is measured by ocular ultrasonography and has been described as an indirect measure of elevated intracranial pressure (ICP) [14,15,16,17]. ONSD is a fast, reliable, low-cost, and non-invasive measure [14,17]. The pressure within the optic nerve sheath has been shown to increase linearly with increasing ICP. ONSD reportedly widens as the ICP increases [17]. In studies conducted on patients with idiopathic intracranial hypertension, psychiatric disorders, including personality disorders, have been found to be quite common [18,19].

Another non-invasive ophthalmologic parameter associated with the nervous system is intraocular pressure (IOP). Studies examining the relationship between psychiatric disorders and IOP are quite limited [9,20]. Moreover, this relationship is mostly attributed to the psychotropic drugs used in treatment [7,21,22]. IOP may be affected by autonomic and hormonal factors. In a study conducted in a non-clinical sample, psychological stress induced in healthy individuals increases IOP by approximately 1 mmHg [23]. Moreover, any history of sexual abuse is associated with a higher risk of primary open-angle glaucoma compared with the absence of such a history [24].

The relationship between dry eye disease (DED), which is another parameter, and psychiatric disorders, has been increasingly recognized as pivotal in recent years. Studies have reported a potential relationship between DED and psychiatric disorders—DED symptoms may be affected by anxiety, depression, schizophrenia, or post-traumatic stress disorder. In addition, psychotropic drugs are also considered risk factors for DED [25,26]. Moreover, a history of CT has been associated with an increased risk of DED in adulthood, especially with DED symptomatology [27].

In summary, the literature has not addressed the correlation between BPD and CT in patients with BPD, on the one hand, and ophthalmologic parameters that can be measured non-invasively, rapidly, and cost-effectively, on the other. The present study aimed to address the aforementioned gaps in the body of literature to compare ophthalmologic parameters, such as ONSD, IOP, and DED, in patients with BPD with healthy controls, and to explore the relationship between these parameters and the severity of CT. Our findings may present new biomarkers and potential indicators of neurobiological changes at the intersection of psychiatry and ophthalmology.

Our study was guided by the following hypotheses:

**H1.** 
*Patients with BPD may have a wider ONSD and higher IOP and DED than healthy controls.*


**H2.** 
*Patients with BPD demonstrate a correlation between specific types of CT and the ophthalmologic parameters of ONSD, IOP, and DED.*


## 2. Materials and Methods

### 2.1. Participants

This study recruited 99 consecutive female patients who applied to the Mental Health and Diseases outpatient clinic of Düziçi State Hospital between 15 November 2024 and 15 May 2025, and met the inclusion criteria. Only female patients were included because BPD is more common in women, and limiting the sample to female patients reduced the potential confounding effect of gender and allowed us to obtain a more homogeneous sample. Consecutive recruitment was preferred to minimize selection bias and to ensure that all eligible patients meeting the strict inclusion and exclusion criteria were systematically enrolled.

This study was designed as an observational, cross-sectional, case-control study, and the reporting process was conducted in accordance with the Strengthening the Reporting of Observational Studies in Epidemiology (STROBE) guidelines.

Inclusion criteria for patients with BPD: We included those aged between 18 and 35 years, with a diagnosis of BPD according to the Structured Clinical Interview for DSM-5 Personality Disorders (SCID-5-PD), with stable symptoms for at least six months, not using psychotropic medication, and who were literate to complete the self-report scale. We included patients who had been diagnosed in the past and still met the diagnostic criteria, even if they were not under active psychopharmacological treatment at the time of the application. Participants had applied to the hospital for routine psychiatric checks, psychoeducation, psychotherapy, or other reasons, such as adaptation difficulties, decreased social functioning, and relationship problems. Notably, BPD, by its nature, makes seeking non-medication support an important part of the treatment process. In addition, we considered patients who were on follow-up without medication as being able to maintain their functionality in the natural course of the disease and fill in the self-report forms correctly, thus ensuring measurement reliability. We included 60 healthy volunteers, who had similar sociodemographic characteristics to the patients, met the inclusion criteria, and routinely visited the ophthalmology outpatient clinic for health screening, as the control group.

Inclusion criteria for healthy controls: The controls needed to be between 18 and 35 years old, not have any mental illness or complaints, and not have a history of psychotropic medication.

We confirmed the absence of a history of mental illness through a psychiatric examination conducted by the first author and review of electronic medical records. In addition, participants were checked for a history of psychotropic drug use through both self-report and medical records.

Exclusion criteria for all participants: We excluded those who reported delirium, dementia, intellectual disability, alcohol or substance use disorders, past recurrent or chronic comorbid mental illnesses, or a history of a systemic or neurological disease.

Ophthalmologic exclusion criteria for all participants: We excluded those who reported the following: any ophthalmologic disease, use of artificial tears in the preceding two weeks, lid abnormalities, history of contact lens use, high refractive error, active ophthalmologic infection, history of orbital–cranial surgery, use of drugs affecting cerebrospinal fluid (CSF) pressure and/or eye examination findings (e.g., tumor, papillary edema, glaucoma), history of cranial and/or orbital radiotherapy, congenital optic disc pathology, smoking, and wide-angle strabismus that prevents visualization of the optic nerve on ultrasonic examination. In addition, female subjects who were menstruating, pregnant, or lactating were excluded from the present study because tear production and stability are significantly related to hormonal changes that occur during menstruation, pregnancy, and lactation [28,29].

The recruited patients with BPD and healthy controls underwent comprehensive psychiatric and ophthalmologic examinations. The clinical interview and ophthalmologic examination of the patients with BPD excluded the following: 20 patients with comorbid major depressive and anxiety disorders, 18 patients with active alcohol–substance use disorder, one patient with regular antihypertensive use for hypertension, one patient with active blepharitis, one patient with contact lens use, two patients with high myopia (≥6 diopters), four patients on their menstrual period who did not re-apply, and two patients who did not volunteer to participate during the evaluation. This study continued with 51 patients with BPD.

As for the healthy controls, the clinical interview and ophthalmologic examination excluded the following: two active smokers, one contact lens wearer, one subject with malpositioned lids, one subject with active conjunctivitis, one subject with a history of orbital surgery, one subject with a high degree of myopia (≥6 diopters), one subject who was breastfeeding, and one subject who did not volunteer to participate in the present study. This study continued with 51 healthy controls (Figure 1).

### 2.2. Procedure

In the first phase of this study, a senior psychiatrist with seven years of experience (first author) interviewed all participants for approximately 30 to 60 min. During these interviews, sociodemographic data of the participants were collected using a form prepared by the authors. The diagnosis of BPD was confirmed using the SCID-5-PD, and other comorbid psychiatric disorders were assessed using the Structured Clinical Interview for DSM-5 Disorders.

We used the Childhood Trauma Questionnaire-Short Form (CTQ-28) to assess traumatic childhood experiences. After the interview, the participants were informed about the scale used. In the next stage, we asked the patients to complete the CTQ-28 to assess traumatic experiences in childhood. For patients who had difficulty completing this self-report scale, the clinician provided assistance and clarified unclear points. In the second stage, the patients were referred to the ophthalmologic outpatient clinic for ophthalmologic examination.

#### Ophthalmologic Examination

All participants underwent a complete ophthalmologic examination, including best corrected visual acuity, applanation tonometry, slit lamp biomicroscopy, and fundoscopy, under the supervision of a senior ophthalmologist (second author) with eight years of experience. All examinations were performed between 9:00 and 13:00. The temperature of the examination room was 25 °C, and the humidity was 50%. None of the patients used eye drops on the day of the examination.

### 2.3. Data Collection

#### 2.3.1. Sociodemographic Data Form

We created a form to gather information on the participants’ age, marital status, duration of education, family history of mental illnesses, and duration of BPD.

#### 2.3.2. Childhood Trauma Questionnaire-Short Form (CTQ-28)

Developed by Bernstein et al. in 2003, the CTQ-28 is a self-report scale that retrospectively assesses traumatic life events experienced by an individual before the age of 20 years [30]. The scale consists of 28 items. Each item has a five-point Likert-type rating from 1 = Never to 5 = Very often. The scale consists of five subscales: sexual abuse (items 20, 21, 23, 24, and 27), physical abuse (items 9, 11, 12, 15, and 17), emotional abuse (items 3, 8, 14, 18, and 25), physical neglect (items 1, 4, 6, 2, and 26), and emotional neglect (items 5, 7, 13, 19, and 28). Each subscale is scored separately. High scores obtained on the scale are associated with the severity of the CT. The CTQ-28 was verified for validity and reliability for Turkish users by Şar et al. in 2012 [31].

#### 2.3.3. Dry Eye Measurement: Tear Break-Up Time (TBUT)

TBUT was measured by the same experienced ophthalmologist. Measurements were performed using sterile fluorescent paper strips (Jinming New Technological Development Co., Ltd., Tongjian, China). A drop of normal saline was put on a strip, which was then gently touched to the inferior temporal bulbar conjunctiva for 1 to 2 s. This was followed by asking the patients to blink naturally several times and then to look straight ahead with their eyes open. The eye was examined under a slit lamp with low magnification and a large light field covering the entire cornea. Under a cobalt blue filter, the time from the last blink to the first appearance of dry spots on the corneal surface was recorded as TBUT. A stopwatch was used to measure the time. The examination was performed three times in each eye to calculate the TBUT, and the mean value of the three measurements was taken as the final TBUT.

#### 2.3.4. Optic Nerve Sheath Diameter (ONSD) Measurement

An experienced ophthalmologist captured B-scan images using transorbital ultrasonography and a linear 7–16 MHz probe (Echoscan US-4000, Nidek Inc., Tokyo, Japan). Participants lay supine on the examination bed with their heads in a comfortable position. A sterile, transparent sticker was used to cover the eyelids and keep the studied eye closed. With the eye closed, the eye socket was filled with water-soluble, conductive gel. The participant was instructed to look with the other eye at the sign on the ceiling. A linear probe was placed in the eyeball filled with gel. The images were captured when the optic nerve sheath was maximally visualized. The measurement was performed 3 mm behind the eyeball on an axis perpendicular to the optic nerve. Three measurements were made for each eye, and the mean value of these three measurements was recorded as the final ONSD of the eye studied.

#### 2.3.5. Intraocular Pressure (IOP) Measurement: Applanation Tonometry

Goldmann applanation tonometry is considered the gold standard among IOP measurement methods. The tonometer was cleaned and dried before each measurement. Anesthetic drops were applied to the eye to be measured. The participant’s head was placed on the biomicroscope, and then the tear film was stained with a fluorescein strip. The light intensity was maximized, and the cobalt blue filter was switched on. The biomicroscope was advanced slowly toward the eye until the tonometer prism was in contact with the participant’s cornea. Subsequently, two half rings of equal size and shape are seen. The tonometer’s drum was rotated until the half rings overlapped at their inner edges—when a horizontal letter S formed. When the letter S was formed, the reading was multiplied by 10 to obtain the IOP in mmHg.

### 2.4. Ethics

The ethics committee approval of this study was obtained at meeting no. 149, dated 8 November 2024, of the Çukurova University Faculty of Medicine, Ethics Committee for Non-Interventional Clinical Research. All participants were informed about this study, and their written consent forms were obtained. Participants were selected on a voluntary basis. This study was conducted in accordance with the principles of the Helsinki Declaration and its subsequent revisions.

### 2.5. Statistical Analysis

We used IBM SPSS Statistics for Windows, version 25, for conducting statistical analyses (IBM Corp., Armonk, NY, USA). The significance threshold was set at *p* < 0.05. The normality of these data was examined using skewness and kurtosis values, and distributions were considered normal if skewness and kurtosis were within the range of ±2 [32]. Sociodemographic variables were compared using chi-square tests, and independent sample *t*-tests were performed to evaluate differences in scale scores and ophthalmological parameters between groups. Pearson correlation analysis was conducted to examine the associations between scale scores and ophthalmological measurements, and these relationships were visualized through a heat map generated in Python (version 3.9) to enhance interpretability.

Furthermore, effect sizes were calculated to evaluate the magnitude of differences. For mean comparisons, Cohen’s *d* was used, and for categorical variables, the φ (phi) coefficient was calculated. According to Cohen, *d* = 0.20 represents a small effect, *d* = 0.50 a medium effect, and *d* = 0.80 a large effect. For φ, 0.10 indicates a small effect, 0.30 a medium effect, and 0.50 a large effect. In correlation analyses, the effect size is directly represented by Pearson’s correlation coefficient (*r*). Cohen’s thresholds for interpreting *r* are as follows: *r* = 0.10 (small), *r* = 0.30 (medium), and *r* = 0.50 (large) [33].

### 2.6. Power Analysis

A post-hoc power analysis was conducted using the G*Power 3.1 software. For independent sample *t*-test comparisons of scale scores between the two groups (51 participants each, total n = 102), the analysis indicated that a statistical power of 80% was achieved with an effect size of 0.5 and a significance level of 0.05. The analyses conducted were one-tailed.

## 3. Results

Table 1 presents the sociodemographic data of the participants. We found no statistically significant difference in terms of age, years of education, marital status, or place of residence between the patient and control groups (*p* > 0.05). The effect size analysis showed that age (d = 0.251) and years of education (d = 0.101) indicated small effects. Both groups showed a significant difference in terms of familial history of mental diseases (*p* < 0.05), with the patient group having more mental diseases in their families. For this variable, the chi-square test yielded a φ coefficient reflecting a large effect (φ = 0.574). In addition, the mean disease duration of the patients who participated in this study was 2.06 ± 2.40 years.

Table 2 provides a comparison of the eye measurement values of the participants between the groups. We found no significant difference between the groups in terms of IOP and DED measurements (*p* > 0.05). The groups showed a significant difference in terms of ONSD measurements (*p* < 0.05). Both the right and left eyes of the patients had larger ONSD measurements compared with the control group. In addition to the assessment of statistical significance, effect sizes were calculated for all comparisons. Examination of the effect sizes revealed that TBUT values showed negative negligible effects (right: d = −0.099, left: d = −0.075), IOP values indicated small effects (right: d = 0.271, left: d = 0.344), and ONSD values demonstrated small-to-moderate effects (right: d = 0.416, left: d = 0.421).

Table 3 shows the total CTQ-28 score and all subscale scores for the two groups. Patients scored higher for the total CTQ-28 measure as well as the emotional abuse, physical abuse, emotional neglect, physical neglect, and sexual abuse subscales compared with the control group (*p* < 0.05). In addition to statistical significance, effect sizes were also calculated for all CTQ subscales and the total score. When the effect sizes were examined, emotional abuse (d = 1.671) and the total CTQ score (d = 1.744) showed very large effects. Emotional neglect (d = 1.066), physical neglect (d = 0.820), and sexual abuse (d = 0.878) demonstrated large effects, whereas physical abuse (d = 0.585) indicated a moderate effect.

We observed a significant positive correlation between the mean score of emotional abuse and the left eye ONSD measurement (r = 0.364, *p* < 0.001, r^2^ = 13.25%), which corresponds to a moderate effect size according to Cohen’s classification. In addition, among the patients, a significant negative correlation was found between the mean score of physical neglect and the right eye IOP measurement (r = –0.287, *p* = 0.04, r^2^ = 8.24%), indicating a small effect size (*p* < 0.05; Table 4). To improve interpretability, we visualized such correlations as a heat map, shown in Figure 2. In the heat map presented in Figure 2, Pearson correlation coefficients between CTQ subscales and ophthalmological variables are visualized. The color scale represents negative correlations in shades of blue and positive correlations in shades of red. The intensity of the color reflects the magnitude of the absolute correlation coefficient. Statistically significant correlations (*p* < 0.05) are indicated with an asterisk (*) in the corresponding cell.

## 4. Discussion

Recent studies aimed at elucidating the biological basis of psychiatric disorders have suggested that ocular structures and functions can be used as potential biomarkers and peripheral indicators of neurobiological changes. The results from these studies reveal how retinal biomarkers can help in making clinical diagnoses and monitoring disease progression [34,35]. The key finding of the present study was that ONSD measurements of both eyes were higher in patients with BPD compared with healthy controls. We also noted a positive correlation between the mean score of emotional abuse and ONSD of the left eye.

The optic nerve is located in a dural sheath that communicates directly with the subarachnoid space. Therefore, the ONSD expands as the ICP increases. Studies examining the relationship between ultrasound-measured ONSD and ICP report that the ONSD is a strong and accurate predictor of the increase in ICP [14,36,37]. Trauma-related audiovisual stimulation may affect CSF concentration in post-traumatic stress disorder [38]. Although BPD is not clearly associated with increased ICP, the observed increase in the ONSD may indicate possible biological changes. Trauma history and pervasive stress, which are common in BPD, may affect brain homeostasis and CSF dynamics. In addition, volumetric neuroimaging research in BPD patients has found a decrease in volume in limbic regions, such as the amygdala and hippocampus [39]. Although this volume reduction is expected to result in a decrease in the ONSD rather than an increase, the ONSD may increase because of the activation of compensatory mechanisms in CSF distribution and neuroinflammatory processes caused by a dysregulated hypothalamic–pituitary–adrenal axis. In other words, an increased ONSD in patients with BPD may reflect a possible change in stress-related ICP. Future neuroimaging and physiological studies are clearly needed to confirm this finding and elucidate possible pathophysiological mechanisms. In addition, considering the prevalence of psychiatric disorders, including personality disorders, in studies involving patients with idiopathic intracranial hypertension [18,19], a reciprocal relationship between increased ICP and BPD is also possible.

The identification of neurobiological biomarkers of traumatic childhood experiences is of great importance for the development of new treatments [40]. In the present study, we found a relationship between emotional abuse and the ONSD of the left eye. Prior research reported that CT may cause permanent microstructural changes in brain regions, such as the limbic system, prefrontal cortex, corpus callosum, and hippocampus [41]. Therefore, stressors such as childhood emotional abuse may leave neuro-ophthalmologic scars and have lasting neurobiological effects that can be measured using peripheral biomarkers. This increase in the ONSD may be an indicator of central nervous system changes that may develop owing to CT. However, longitudinal studies including biochemical parameters are needed to confirm this relationship.

A recent study reported that CT does not have any significant effect on the response abnormalities of retinal neurons (cone and rod cells) in children and adolescents at familial risk for psychosis or mood disorders. However, a specific type of trauma (physical violence against the child) may have a specific effect on the prolonged latencies of cone and rod cells in girls [10]. In our study, we found a significant relationship between another specific type of trauma (emotional abuse) and the ONSD. This finding is coherent with the view that different types of CT may have different effects on the brain and different neurobiological systems [41,42]. The moderate effect size of this association suggests that emotional abuse, in particular, may be linked to neurobiological alterations, consistent with evidence that CT can differentially affect brain systems.

Acute psychological stress can increase IOP in healthy individuals [23]. In our study, patients with BPD did not show a significant increase in IOP compared with healthy controls. Although this finding may seem contradictory at first, it makes sense given that the individuals in our sample had not taken psychotropic drugs in the preceding six months and were included in this study when they were in a stable period (they did not show any active psychotic or dissociative crisis, agitation, suicidal tendency, or any other similar acute clinical finding). Meanwhile, research has previously shown that the possible increase in IOP levels may be caused by antipsychotics rather than the disease itself. A study of patients with schizophrenia under medication reported IOP abnormalities in patients using ziprasidone [20]. Another study comparing the optical coherence tomography (OCT) findings of patients with BPD who had not used psychotropic drugs for at least four months with a control group found no significant difference in terms of IOP measurements of the groups, which was consistent with our study [9]. In the present study, we took psychotropic drug use and comorbid psychiatric disorders that may affect IOP as exclusion criteria. Given that the confounding factors affecting IOP were controlled in both studies, the possible IOP changes in patients with BPD may be caused by the medications used and comorbid psychiatric disorders rather than the disease itself. Moreover, the effects on IOP may rather be associated with the psychophysiologic burden in the acute phase of the disease, leading to the absence of any significant difference in the measurements performed during stable periods.

In the present study, we found no relationship between the IOP and specific CTs, such as physical abuse, emotional neglect, emotional abuse, or sexual abuse. In contrast, Yu et al. found a significant correlation between a history of sexual abuse at an early age and primary open-angle glaucoma [24]. However, these two studies yielding different results can be explained by the significant methodological differences in assessment methods and sample characteristics. First, in our study, we measured IOP using Goldmann applanation tonometry. We made no diagnosis of glaucoma and evaluated only momentary physiological pressure values. Meanwhile, Yu et al. examined individuals diagnosed with glaucoma, a diagnosis that is made by evaluating structural and functional criteria related to visual field loss and optic nerve damage together, rather than IOP values alone. Therefore, a direct correspondence between the diagnosis of glaucoma and IOP measurement is not expected, and glaucoma can be diagnosed even when IOP is within normal limits, as in normotensive glaucoma cases. In addition, the sample in Yu et al. consisted of women in an older age group (40 years and older) and may reflect the subsequent effects of structural degenerative processes that may develop after the trauma. In our study, we evaluated younger (18–35 years old) individuals with BPD who were stable for at least six months and who were not taking medication. Potential trauma-related changes in the IOP in these younger individuals may not have yet evolved into glaucoma-like structural impairments. In addition, the way trauma is defined, the method of measurement (self-report vs. diagnosis-based records), the age period of exposure, and differences in comorbid psychiatric conditions are also key factors that explain the disparity in the results of the two studies. For these reasons, although both studies are based on the same theoretical framework, their results cannot be expected to directly overlap, as they differ significantly in terms of the biological parameters, sample characteristics, and clinical processes being evaluated. This does not reduce the significance of the negative findings in our study; on the contrary, it suggests that changes in the IOP may vary over time, depending on the disease process and clinical context.

Another notable finding in the present study was the negative correlation between physical neglect scores and right eye IOP. However, since this correlation had a small effect size and is not supported by former studies, the possibility that this finding may have occurred by chance should not be ruled out. Therefore, further studies with larger sample sizes are needed to elucidate the possible effects of physical neglect on IOP.

Different objective measurement tools, such as the Schirmer test, corneal fluorescein staining, and TBUT, have been used to investigate the relationship between DED and psychiatric disorders [43]. In our study, we compared the patient and control groups in terms of DED using TBUT, finding no significant difference between the groups. Kitazawa et al. found that the severities of depression and anxiety are associated with subjective DED complaints but not with objective examination findings [44]. Similarly, our study’s objective tests did not yield any significant difference in other psychiatric disorders. However, given that we evaluated only objective parameters, we could not interpret whether subjective complaints in patients with BPD overlap with objective findings. In addition, as in the case with IOP, DED findings may be related to the medications used and comorbid psychiatric disorders rather than the disease itself. However, studies with newly diagnosed, drug-naive patients with OCD and newly diagnosed patients with depressive disorder have also shown a relationship between these disorders and DED [45,46]. More comprehensive studies with controlled drug use and comorbidities are needed to elucidate the issue specifically in the context of BPD.

Our study also found no relationship between CTs and DED. Studies with large sample groups of healthy individuals have reported an associated between psychological stress and DED [47,48]. In healthy individuals, a history of CT is associated with an increased risk of DED in adulthood and specifically with DED symptoms [27]. In prior studies, DED was assessed using self-report scales, whereas in our study, DED was assessed by eye examination using TBUT. These differences in results may be attributed to the differences in measurement tools and sample size. In addition, prior studies have generally examined individuals without a psychiatric diagnosis. Our study included patients with BPD, who presented a different clinical profile in this respect. Given that the nature of psychiatric disorders may alter the direction and severity of physiological responses, different physiological outcomes may be observed despite similar trauma histories. Therefore, comparative studies with similar instruments and larger samples are needed to elucidate the relationship between DED and CT in specific diagnoses, such as BPD.

Our study complements the recent OCT research in patients with BPD. Xu et al. found that the retinal nerve fiber layer and central retina of patients with BPD are thinner compared with healthy controls [9]. When these ocular parameters (ONSD measured by ultrasound and retinal measurements by OCT) are evaluated together, they paint a picture of the brain–eye axis being affected more extensively in BPD, further supporting the concept of ocular measurements as objective indicators of neurobiological changes.

In our study, consistent with the body of literature, all CTQ-28 subscales and total scores of the patients with BPD were significantly higher compared with the healthy controls. In their meta-analysis, Porter et al. found that patients with BPD are 13 times more likely to report adverse life events in childhood compared with non-clinical controls. They also reported a higher likelihood of reporting adverse life events experienced during childhood compared with the clinical population (other psychiatric patients) [49].

As expected, in the present study, a family history of mental illness was more common in patients with BPD than in healthy controls. A genetic predisposition for BPD has been found in family and twin studies [50]. Skoglund et al. reported that BPD clustered in families and that heritability was estimated to be 46% [51].

### 4.1. Strengths

To the best of our knowledge, our study is the first to demonstrate an increase in the ONSD and its relationship with CT in patients with BPD. The inclusion of patients not taking medication for at least six months and the exclusion of patients with comorbid psychiatric disorders prevented the possible effects of confounding factors on these measurements.

### 4.2. Limitations

Our study had some limitations. First, the cross-sectional design of this study prevents causal inference. Second, the relatively small sample size and the fact that this study was conducted in a single center and consisted exclusively of women may limit the generalizability of this study. In addition, the limited sample size prevented subgroup and multivariate analyses. Third, ICP measurements or neuroimaging methods were not used to confirm the ONSD findings. Fourth, cortisol levels, inflammatory markers, or autonomic functions were not measured. Therefore, possible links in pathophysiology could not be fully confirmed. Last, although the measurements were performed by an experienced ophthalmologist, the doctor’s prior knowledge of the patients and healthy controls may have consciously or unconsciously influenced the measurements, leading to measurement bias. The psychiatric assessment of participants was conducted using a Structured Clinical Interview and the SCID-5-PD module. Therefore, the risk of bias in diagnosis and group allocation was considered minimal.

Further studies with longitudinal designs in larger sample groups, validated by neuroimaging methods and inflammatory biomarkers, are needed to support our findings. Future studies should examine whether trauma-focused therapies or stress management can normalize these ophthalmologic measurements. Such studies could pave the way for eye-based biomarkers to gain prominence in the diagnosis and management of psychiatric disorders associated with severe stress. Concurrent neuroimaging studies could clarify whether the increase in ONSD corresponds to changes in brain volume. It would also be valuable to examine the relationship of such ophthalmologic parameters with other clinical variables, such as suicidality, self-mutilation, and impulsive behavior. In summary, future studies should investigate the neurobiological underpinnings of our findings and test their relevance for a better understanding and management of BPD. Further research in this direction could provide long-needed objective indicators in psychiatry, creating new paths for diagnosis and treatment.

## 5. Conclusions

We found that patients with BPD had a higher ONSD compared with healthy controls, and that this may be related to specific types of CT, such as emotional abuse. These findings are both clinically and theoretically relevant. Clinically, ONSD may serve as a promising peripheral biomarker and objective indicator of neurobiological changes in BPD and help elucidate the neurobiological basis of BPD. In addition, non-invasive ophthalmologic indicators, such as ONSD, may serve as potential tools for assessing stress levels in BPD, especially in patients with a history of CT. From a theoretical standpoint, these findings can be considered to support the biopsychosocial nature of the disease by associating CT with measurable physiological changes and underlining the embodiment of psychological traumas. Our results emphasize the importance of addressing CT in treatment.

Our study also emphasizes the need for holistic care. Particularly in complex cases, neurological or ophthalmological evaluation can complement the psychological assessment. If future studies confirm the association between patients with BPD with a history of CT and increased ONSD, clinicians may consider incorporating ocular consultation in their routine practice to perform ocular ultrasound, a rapid and non-invasive method for high-risk patients. Thus, ophthalmological evaluations can be part of comprehensive care in patients with BPD.

## Figures and Tables

**Figure 1 jcm-14-05886-f001:**
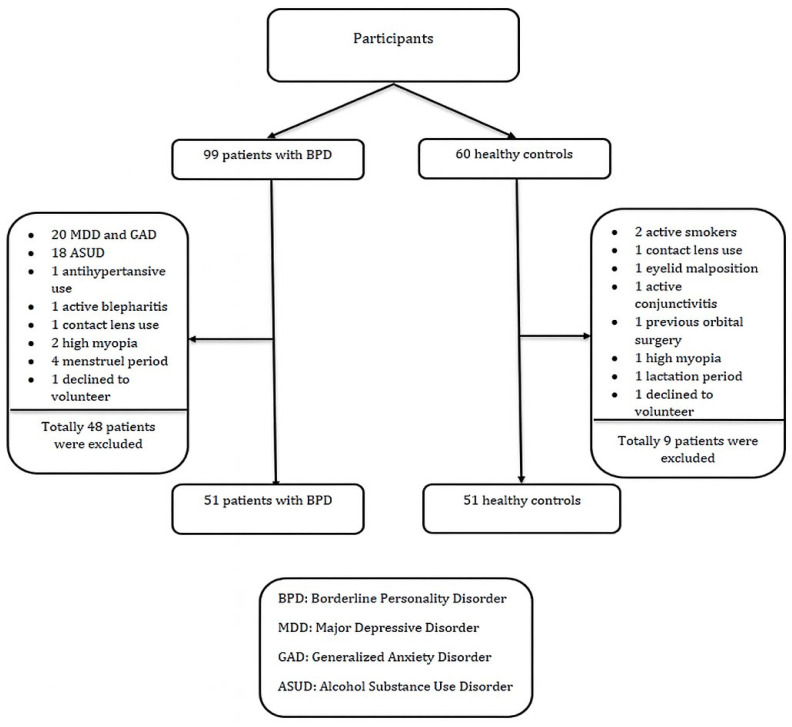
Flowchart illustrating the inclusion and exclusion process of participants.

**Figure 2 jcm-14-05886-f002:**
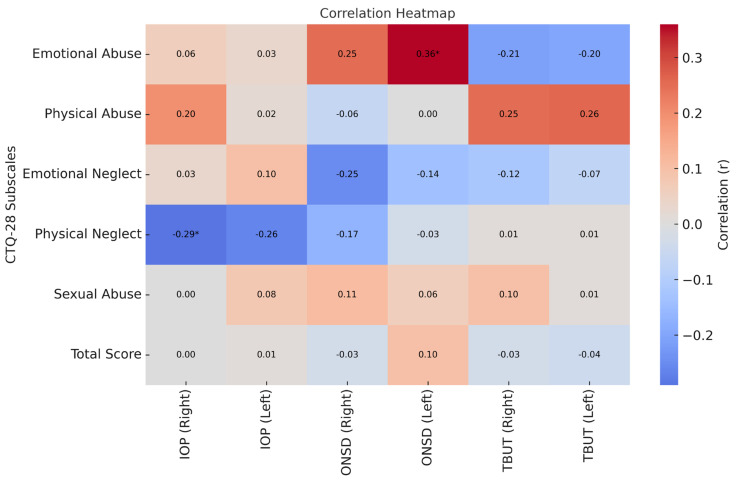
Heat map showing the correlations between CTQ-28 subscales and ophthalmological parameters. Red indicates positive correlations, while blue indicates negative correlations. The intensity of the color corresponds to the strength of the correlation. Only statistically significant correlations (*p* < 0.05) are marked with an asterisk (*).

**Table 1 jcm-14-05886-t001:** Comparison of sociodemographic characteristics of groups.

Descriptive Characteristic	BPD		Control		Effect Size	*p*-Value
	Number (n)	Percentage (%)	Number (n)	Percentage (%)		
Marital status						
Single	43	84.3	45	88.2		*p* > 0.05 *
Married	6	11.8	6	11.8		
Divorced	2	3.9	0			
Place of residence						
Provincial center	14	27.5	9	17.6		*p* > 0.05
Smaller units than a provincial center	37	72.5	42	82.4		
Mental illness in the family Yes No					(φ)	
	32	62.7	4	7.8		***p* < 0.05**
	19	37.3	47	92.2	0.574	
Ages (years)	Mean ± SD 21.13 ± 3.92		Mean ± SD 20.03 ± 4.46		(d) 0.251	*p* > 0.05 **
Years of education	Mean ± SD 12.78 ± 2.30		Mean ± SD 12.58 ± 1.13		(d) 0.101	*p* > 0.05 **
Years of illness	Mean ± SD 2.06 ± 2.40					

*: Fisher’s exact chi-squared test, **: independent *t*-test, BPD: Borderline personality disorder, SD: Standard deviation, (d): Cohen’s d (effect size for mean comparisons), (φ): phi coefficient (effect size for chi-square tests). Statistically significant differences are highlighted in bold.

**Table 2 jcm-14-05886-t002:** Comparison of ophthalmologic parameters of the groups.

	BPD Mean ± SD	Control Mean ± SD	t	Effect Size (d)	*p*-Value
IOP-right (mmHg)	16.45 ± 3.98	15.52 ± 2.70	t: 1.36	0.271	*p* > 0.05
IOP-left (mmHg)	17.94 ± 9.54	15.52 ± 2.68	t: 1.73	0.344	*p* > 0.05
ONSD-right (mm)	3.97 ± 0.47	3.78 ± 0.45	t: 2.10	0.416	***p* < 0.05**
ONSD-left (mm)	3.94 ± 0.43	3.76 ± 0.44	t: 2.12	0.421	***p* < 0.05**
TBUT-right (seconds)	7.74 ± 4.09	8.11 ± 3.39	t: −0.50	−0.099	*p* > 0.05
TBUT-left (seconds)	7.86 ± 3.67	8.15 ± 4.10	t: −0.38	−0.075	*p* > 0.05

t: independent *t*-test, BPD: Borderline personality disorder, IOP: Intraocular pressure, ONSD: Optic nerve sheath diameter, TBUT: Tear break-up time, SD: Standard deviation, (d): Cohen’s d (effect size for mean comparisons). Statistically significant differences are highlighted in bold.

**Table 3 jcm-14-05886-t003:** Comparison of CTQ-28 scale scores of the groups.

		BPD (n = 51) Mean ± SD	Control (n = 51) Mean ± SD	t	Effect Size (d)	*p*-Value
CTQ-28	Emotional Abuse	11.37 ± 4.41	5.90 ± 1.38	t: 8.43	1.671	***p* < 0.001**
	Physical Abuse	6.43 ± 2.77	5.23 ± 0.81	t: 2.95	0.585	***p* < 0.001**
	Emotional Neglect	14.33 ± 4.58	9.49 ± 4.49	t: 5.38	1.066	***p* < 0.001**
	Physical Neglect	9.11 ± 3.22	6.94 ± 1.91	t: 4.14	0.820	***p* < 0.001**
	Sexual Abuse	7.52 ± 4.07	5.00 ± 0.00	t: 4.43	0.878	***p* < 0.001**
	Total Score	48.78 ± 11.49	32.56 ± 6.38	t: 8.80	1.744	***p* < 0.001**

t: independent *t*-test, BPD: Borderline personality disorder, CTQ-28: Childhood Trauma Questionnaire-Short Form, SD: Standard deviation, (d): Cohen’s d (effect size for mean comparisons). Statistically significant differences are highlighted in bold.

**Table 4 jcm-14-05886-t004:** Relationships between ophthalmologic parameters and CTQ-28 scores of patients with BPD.

			IOP		ONSD		TBUT	
			Right	Left	Right	Left	Right	Left
CTQ-28	Emotional Abuse	r	0.055	0.030	0.25	**0.364** **	−0.209	−0.200
		*p*	0.70	0.83	0.07	***p*** **< 0.001**	0.14	0.16
	Physical Abuse	r	0.201	0.021	−0.055	0.002	0.249	0.257
		*p*	0.15	0.88	0.70	0.98	0.07	0.62
	Emotional Neglect	r	0.031	0.100	−0.251	−0.144	−0.125	−0.070
		*p*	0.82	0.48	0.07	0.31	0.38	0.62
	Physical Neglect	r	**−0.287** *	−0.264	−0.170	−0.034	0.007	0.005
		*p*	**0.04**	0.06	0.23	0.81	0.98	0.97
	Sexual Abuse	r	0.001	0.077	0.107	0.063	0.097	0.012
		*p*	0.99	0.58	0.45	0.65	0.49	0.93
	Total Score	r	0.002	0.010	−0.027	0.096	−0.034	−0.037
		*p*	0.09	0.94	0.85	0.50	0.81	0.79

Pearson correlation analysis *: *p* < 0.05, **: *p* < 0.001, CTQ-28: Childhood Trauma Questionnaire-Short Form, IOP: Intraocular pressure, ONSD: Optic nerve sheath diameter, TBUT: Tear break-up time. Statistically significant differences are highlighted in bold.

## Data Availability

Data are available on request from the corresponding author. These data are not publicly available due to ethical restrictions and privacy concerns involving human participants.

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
