# Peer review of "Investigation of Optic Nerve Sheath Diameter, Intraocular Pressure, and Dry Eye in Patients with Borderline Personality Disorder: The Role of Childhood Trauma"

_jcm, 2025, doi:10.3390/jcm14165886_

Round 1
Reviewer 1 Report
Comments and Suggestions for Authors
The authors have done a great job of describing in detail step by step approach they took to conduct the study.
A couple of clarifying responses from the authors will be helpful to greater understanding of the study methods and results.
- Did the authors conduct a power analysis to determine recommended sample size? If so, it should be stated and discussed.
- Did the authors examine effect size and can they comment on the magnitude of the correlations observed and implications of the results?
- In the results section, provide direction/guidance on how to read and interpret the Heat Map.
- In the limitations section, the authors note that the opthalmologist's prior knowledge of the study subjects may have may have "consciously or unconsciously influenced the measurements, leading to measurement bias" (p. 13). The authors should comment on whether this would be also applicable to sample selection, specific to screening, assessment, and diagnosis of the psychiatric conditions/ Borderline Personality Disorder in determining treatment vs. control group.
Reviewer 2 Report
Comments and Suggestions for Authors
- What is the main question addressed by the research?
This study investigates the relationship between non-invasive ophthalmological parameters [i.e. optic nerve sheath diameter (ONSD), intraocular pressure (IOP), and dry eye disease (DED)], and childhood trauma (CT) in patients with borderline personality disorder (BPD) compared to healthy controls.
- Do you consider the topic original or relevant to the field? Does it address a specific gap in the field? Please also explain why this is/ is not the case.
To the best of my knowledge there is scarce literature about the non-invasive ophthalmological parameters and borderline personality disorder. There is evidence in two systematic reviews/meta-analyses that mental disorders have a potential correlation with childhood trauma, but there is no evidence about the aforementioned non-invasive parameters.
- What does it add to the subject area compared with other published material?
This study offers new data about the non-invasive ophthalmological parameters and their prognostic or diagnostic potential, in patients with BPD and childhood trauma. The results may offer a springboard for new research in diagnosis and/or prognosis BPD in the context of childhood trauma.
- What specific improvements should the authors consider regarding the methodology?
- Please correct the first sentence of section 2.1. “The study recruited 99 consecutive female patients…criteria”. Does this mean that only female patients were recruited? If so, this needs explanation why you chose to include only female patients, and why they were consecutive.
- You could state the type of study (e.g. prospective, observational, case-control, cohort, or other), and any guideline that was followed, e.g. STROBE for observational studies.
- It would be to see the recruitment period and how the participants were recruited e.g. random sampling or other.
- Have you done an a priori power calculation to justify the sample size? If no, it would be nice to see an explanation or you could include it in the limitations.
- I would also recommend justifying the choice of statistical tests. Have you considered any normality tests?
- Are the conclusions consistent with the evidence and arguments presented and do they address the main question posed? Please also explain why this is/is not the case.
At the end of section 1, it is stated that “The study was guided by the following hypotheses… DED”. The conclusion is consistent with the main research question, where the authors stated that patients with BPD had a higher ONSD compared with healthy controls, and that this may be related to specific types of CT, such as emotional abuse.
- Are the references appropriate?
Yes, they are
- Any additional comments on the tables and figures.
Please check all the tables for re-alignment. Please keep a uniform appearance regarding p value, e.g. please consider changing p =0.00 in Table 3, into p < 0.001, if that is the case. Please consider a legend explaining the color scale and indicating the significance threshold.
